# Insect-Specific Viruses and Their Emerging Role in Plant Disease Mitigation

**DOI:** 10.3390/v17091269

**Published:** 2025-09-19

**Authors:** Jianing Lei, Jingna Yuan, Mengnan Chen, Qianzhuo Mao

**Affiliations:** State Key Laboratory for Quality and Safety of Agro-Products, Key Laboratory of Biotechnology in Plant Protection of MARA, Key Laboratory of Green Plant Protection of Zhejiang Province, Institute of Plant Virology, Ningbo University, Ningbo 315211, China; ljn13849163355@163.com (J.L.); yjn2430304122@163.com (J.Y.); 15066716480@163.com (M.C.)

**Keywords:** insect-specific virus, plant virus, insect vector, virus transmission

## Abstract

Insect vectors play a pivotal role in the emergence and dissemination of plant viral diseases. Beyond their function in transmitting plant viruses, these insects harbor diverse insect-specific viruses (ISVs). Advances in high-throughput sequencing (HTS) have uncovered virus diversity and prevalence in insects that far exceed previous estimations. However, current knowledge of ISVs remains predominantly limited to genomic sequencing information. Investigating the fundamental biology of ISVs, their effects on insect physiology, and their modulation of vector competence is critical for deciphering complex virus–virus and virus–insect interactions. Such research holds substantial promise for developing innovative biocontrol strategies against plant viral pathogens. This review synthesizes current insights into the interplay between plant viruses and their insect vectors, explores the discovery and functional roles of ISVs, and discusses the potential application of ISVs in mitigating plant viral diseases. Understanding these dynamic relationships offers new avenues for sustainable plant disease management.

## 1. Introduction

Plant viruses pose a significant threat to crop yield and quality, substantially hampering agricultural productivity [1,2,3]. They manifest in various forms and are spread through different mechanisms. Of these, approximately 80% of known plant viruses are transmitted by insects [4]. Before transmission from insects to plants, viruses interact with various tissues of their insect vectors [3,4,5]. As a result, the interplay between insect population dynamics and the effectiveness of viral transmission by vectors crucially influences the severity of viral disease outbreaks. Conventional control strategies predominantly depend on pesticides or insecticides to eliminate vector insects, thereby constraining the dissemination of plant viruses [5,6].

Insects not only serve as vectors for transmitting plant, animal, and human viruses but also act as hosts to a wide range of viruses [6,7,8]. Additionally, viruses present in insects may arise from their diet, habitat, or symbiotic microbiota. The emergence and utilization of high-throughput sequencing (HTS) technologies have unveiled a considerable number of virus sequences in insects. These viruses are commonly known as insect-specific viruses, insect symbiotic viruses, or insect-associated viruses, reflecting their primary infectivity to insects or close association with them [6,9,10]. Throughout this paper, the term “insect-specific viruses (ISVs)” will be consistently employed. Recent years have witnessed a considerable viral diversity phytophagous insects like whiteflies, leafhoppers, planthoppers, grasshoppers, thrips, aphids, and beetles [9,11,12,13,14,15,16,17]. The continuous discovery of new viruses has intensified interest in their functional roles. Studies reveal that viruses not only induce pathogenic effects but also influence critical biological processes in insects, spanning growth, development, sexual differentiation, fecundity, and vector competence, resulting in outcomes that may range from beneficial to detrimental [7,18].

The efficacy of plant virus transmission differs among various insect populations. ISVs represent a significant category of symbiotic microorganisms in insects, and the diversity of viruses in insects is influenced by factors such as population, season, weather conditions, and geographical location [11,13,18]. These newly identified viral sequences provide exceptional prospects for investigating virus–host coevolution and interactions, with potential implications for manipulating insect traits. Given that most ISVs specifically infect insects, they serve as promising tools for managing both insect vectors and the pathogens they transmit—whether utilized as wild-type agents or engineered recombinants. This review focuses on ISVs, examining the interplay between plant viruses and vector insects, the discovery and functional roles of ISVs, and the potential application of ISVs in mitigating plant viral infections.

## 2. Molecular Mechanisms of Plant Virus Transmission by Insect Vectors

The long-distance migration of vector insects facilitates the spread of viruses, leading to epidemic outbreaks and significantly complicating disease management. Approximately 55% of plant viruses are transmitted by vectors with piercing-sucking mouthparts, primarily aphids, planthoppers, leafhoppers, whiteflies, and thrips [1,2,3]. Plant viruses are categorized into two transmission modes based on their pathways within insect vectors: non-circulative and circulative [1,2,4]. Non-circulative viruses, which include non-persistent and semi-persistent types, briefly attach to the cuticular surfaces of the vector′s mouthparts (stylets) or foreguts [5,6]. This temporary attachment allows for swift acquisition of virions during plant feeding and immediate inoculation into subsequent host plants. An illustrative instance is the cauliflower mosaic virus (CaMV, *Caulimovirus tessellobrassicae*), utilizing the helper protein P2 to specifically bind to the acrostyle region in aphid mouthparts for efficient non-circulative transmission [7,8].

Viruses demonstrating persistent circulative transmission establish enduring relationships with their insect vectors. Following acquisition during plant feeding, these viruses traverse the insect′s digestive system, disseminate systemically, and eventually locate the salivary glands. Subsequent salivation during feeding on new host plants facilitates viral inoculation [1,3,4]. Within this category, persistent nonpropagative viruses complete this cycle without replicating in the vector, unlike persistent propagative viruses that actively replicate within their insect hosts. This replication process effectively designates the vector as an intermediate reservoir host in natural transmission cycles. Some plant viruses can achieve vertical transmission within insect populations, ensuring their environmental persistence irrespective of plant hosts [9]. The identified persistent propagative viruses primarily belong to the *Bunyavirales*, *Rhabdoviridae*, and *Reovirales*—taxonomic groups that also encompass numerous animal-infecting viruses [6]. This correlation hints at close evolutionary connections between plant-infecting arboviruses (phytoarboviruses) and their animal-infecting counterparts.

During the process of infection and circulation within insect hosts, persistent propagative viruses must overcome multiple tissue and membrane barriers and also evade the insect′s immune system to replicate within cells [10,11]. This process has been extensively researched and reported over the past decade. These complex processes have been extensively characterized over the past decades. A prime example is Rice dwarf virus (RDV, *Phytoreovirus alphaoryzae*), a reovirus transmitted by the leafhopper *Nephotettix cincticeps*. Upon ingestion, RDV first establishes primary infection foci in epithelial cells of the filter chamber within the digestive tract. Utilizing specialized viroplasms as replication and assembly sites, the virus employs its nonstructural protein Pns10 to form tubules for moving virions through microvilli into the midgut epithelium. Further dissemination unfolds via muscular tissues and hemolymph, ultimately reaching the salivary glands [12,13,14,15]. In leafhoppers, cell entry relies on the minor capsid protein P2, crucial for successful infection. Notably, prolonged maintenance of RDV in rice plants leads to nonsense mutations in the S2 gene segment, abolishing P2 expression and consequently vector transmissibility [16]. This functional degeneration provides evolutionary evidence that ancestral RDV likely originated as an insect-specific virus prior to adapting to plant hosts.

The insect innate immune system encompasses defenses rooted in the digestive tract, cellular responses, as well as humoral immunity, RNA interference, and crucial signaling pathways such as the Toll, Imd, and JAK/STAT pathways. Autophagy and apoptosis also play essential roles [17,18]. These innate immune barriers ensure insect health and impact the interactions between vectors and viruses. In response, viruses have evolved intricate strategies to inhibit or utilize insect immune reactions during circulation, promoting survival, replication, and efficient transmission. For example, tomato yellow leaf curl virus (TYLCV, *Begomovirus coheni*) coat protein manipulates phosphatidylethanolamine-binding protein 4 (PEBP4) on whitefly (*Bemisia tabaci*) midgut and salivary gland membranes, activating apoptosis and autophagy pathways to establish a balanced immune response that supports both vector survival and viral transmission [19]. The Toll–Dorsal immune axis in the small brown planthopper (*Laodelphax striatellus*) provides antiviral defense. Conversely, rice stripe virus (RSV, *Tenuivirus oryzaclavatae*), transmitted by this insect, utilizes its non-structural protein NS4 to inhibit dorsal phosphorylation, thereby attenuating the host′s antiviral response [20].

During transmission by insect vectors, plant viruses interact directly with insects, reshaping their immune and nervous systems to affect host choices, feeding habits, and reproduction. This interference influences viral transmission efficiency by insects. An illustrative case is tomato spotted wilt virus (TSWV, *Orthotospovirus tomatomaculae*), which endures persistent propagative transmission through thrips vectors. Infected female thrips exhibit extended copulation and enhanced egg production [21]. Meanwhile, infected males prolong feeding times and increase probing frequency threefold compared to uninfected counterparts [22]. In sum, these virus-induced behavioral alterations amplify TSWV dissemination by thrips.

## 3. Discovery and Limitations of ISVs Research

Recent advancements in HTS and analytical tools have facilitated the large-scale recognition of novel viral sequences. Specifically, ISVs have emerged as a significant proportion of these discoveries. Piercing-sucking insects from the orders Hemiptera and Thysanoptera serve as vectors for transmitting more than half of all known plant viruses [1,2,3]. In this study, we focused on piercing-sucking insects from the Hemipteran families Aphididae, Aleyrodidae, Psyllidae, Cicadellidae, and Delphacidae, as well as the Thysanopteran family Thripidae, and compiled statistics on the number, classification, and related information of viruses reported in the NCBI Virus Database. The NCBI Virus Database (https://www.ncbi.nlm.nih.gov/labs/virus) (accessed on 7 August 2025) is a major, frequently updated community portal that integrates viral sequence data from RefSeq, GenBank, and other NCBI repositories, and serves as a primary reference resource despite limitations such as incomplete genome sequences or uncertain host information [23,24]. Over the past decade (2016–present), numerous novel viruses have been discovered and reported annually. As of August 2025, a total of 441 viruses have been documented in 74 species of piercing-sucking insects (Figure 1A,B). At the family level, these viruses are predominantly classified into *Iflaviridae*, *Dicistroviridae*, *Phenuiviridae*, *Flaviviridae*, *Partitiviridae*, *Rhabdoviridae*, *Geminiviridae*, *Orthomyxoviridae*, *Parvoviridae*, and *Chuviridae*, with approximately 140 viruses remaining unassigned to a definitive taxonomic category (Figure 1C). This diversity highlights the remarkable variety of viruses harbored by insect hosts.

HTS platforms enable a thorough profiling of complex genetic mixtures, enhancing the sensitive detection of both DNA and RNA viruses regardless of the viral load in host tissues [25,26,27]. Various HTS sampling techniques, library preparation methods, and bioinformatics analyses have been employed in the identification of viral sequences across a range of host tissues. Each methodology possesses distinct advantages and limitations. Notably, transcriptome sequencing and small RNA sequencing persist as the most commonly utilized methodologies for virome exploration [28]. Transcriptome sequencing employs HTS to scrutinize the complete set of cellular transcripts. However, this approach has inherent limitations: contigs exhibiting viral sequence homology may lead to false-positive identifications, particularly with endogenous viral elements (EVEs), which refer to viral sequences integrated into host genomes [29,30]. Additionally, RNA-Seq cannot definitively distinguish whether the identified transcripts originate from the insect host itself or from potential contaminants such as symbiotic microbiota (e.g., gut microbes or parasite-associated viruses), dietary components (plant/animal-derived materials), or environmental sources [25,31]. Consequently, transcriptome sequencing is frequently complemented with small RNA (sRNA) profiling [25,28,32]. Viral infections stimulate the production of distinct small RNA species that reflect the infecting virus. In arthropods, the primary antiviral defense mechanism operates through the siRNA pathway, where viral double-stranded RNA is cleaved into 21–23 nucleotide siRNAs [18,33,34]. Thus, virus-derived small RNAs offer crucial evidence to authenticate viral infections during pathogen discovery.

Conventional bioinformatic approaches for identifying viral sequences in HTS data primarily rely on sequence homology searches. However, this approach often fails to detect viruses lacking similarity to sequences found in known genetic databases [35]. An illustrative case of this shortcoming is the discovery of Quenyaviruses. These agents were initially found in *Drosophila melanogaster* based on distinct small RNA profiles. The virus-derived small interfering RNAs (vsiRNAs) exhibited a characteristic size class of 21 nucleotides [36,37]. Importantly, these vsiRNAs did not show homology with any documented viral or cellular genomes. The incorporation of deep learning algorithms into metagenomic analysis marks a significant development, displaying immense potential in expanding virus discovery capabilities beyond traditional homology-based methods.

The utilization of HTS has refined viral classification procedures, allowing the International Committee on Taxonomy of Viruses (ICTV) to assign novel viruses categories based solely on complete RNA-dependent RNA polymerase (RdRP) sequence delineations [27,38]. A case in point is *Jingchuvirales*, which were first observed in arthropod metagenomic studies as “chuviruses” and tentatively grouped within the *Mononegavirales* order using phylogenetic analyses [39,40,41]. Further investigations unveiled distinctive genomic structures and evolutionary differences, leading to the official recognition of *Jingchuvirales* as a separate viral order in 2022 [39,42]. Nevertheless, current knowledge remains confined to sequence data, leaving essential questions unanswered about their natural forms, survival mechanisms, virion configurations, host ranges, and intrahost transmission-replication strategies.

## 4. ISVs Shaping Insect Biology and Virus Transmission

### 4.1. Physiological Modulation of Insects by ISVs

The initial discovery of viruses in insects primarily involved pathogenic viruses, such as baculoviruses [31]. Certain baculovirus species are currently utilized as biopesticides to manage specific Lepidopteran pests, with some products commercially available [43]. With the accelerated discovery of ISVs, their multifunctional impacts on host biology have become increasingly evident. ISVs manifest a variety of effects on crucial physiological processes in insect hosts, encompassing reproduction, development, survival, sex ratio regulation, and immune modulation.

Antagonistic relationships occur when ISVs impose detrimental effects on hosts, including lethal consequences. For instance, research has demonstrated that a baculovirus, Lymantria dispar multiple nucleopolyhedrovirus (LdMNPV, *Alphabaculovirus lydisparis*), infecting Lymantria dispar larvae triggers climbing behavior in the infected larvae. Normally, *L. dispar* caterpillars reside in the soil and emerge solely for nocturnal feeding to evade predators. However, infected larvae exhibit a contrasting behavior by climbing up trees before perishing, thereby dispersing the enclosed virus onto the plants beneath them [43]. Moreover, Pteromalus puparum negative-strand RNA virus 1 (PpNSRV1, *Peropuvirus pteromali*) induces female-biased offspring mortality, skewing sex ratios and diminishing reproductive fitness [44]. Kallithea virus (*Alphanudivirus dromelanogasteris*) exhibits significant pathogenicity in adult *D. melanogaster*, achieving comparable viral titers in both sexes. Infection substantially reduces male survival rates while impeding female mobility and late-stage oviposition capacity [45]. Similarly, the Spodoptera litura male-killing virus (SlMKV) selectively eradicates male offspring in *Spodoptera litura* [46].

Mutualistic symbiosis characterizes interactions in which ISVs provide adaptive advantages to their hosts. For example, parasitoid wasps lay their eggs within live insect larvae, triggering the host′s innate immune system to encapsulate the eggs in melanized capsules, halting embryonic development. However, a parasitoid wasp-specific virus can suppress this encapsulation response, ensuring the survival of the eggs [47,48,49]. Recent research indicates that Rondani’s wasp virus 1, a novel cripavirus associated with parasitoids, infects the host *D. melanogaster*. This infection prolongs the host′s development, increases the population density of flies, and secures resources for the developing wasp offspring [50]. In the leafhopper *Recilia dorsalis*, a symbiotic virus assigned to *Virgaviridae* extends the longevity of males, accelerates oocyte maturation, and boosts female fecundity by around 20% [51]. Similarly, the Acyrthosiphon pisum virus (APV) enhances the adaptation of pea aphids (*Acyrthosiphon pisum*) to suboptimal host plants by modulating the plants′ phytohormone defenses [52].

In reality, it is difficult to characterize the relationship between ISVs and their hosts merely as uniformly beneficial or detrimental. The same virus can be beneficial under specific conditions while imposing fitness costs in others. For instance, the DNA virus Dysaphis plantaginea densovirus 1 (*Hemiambidensovirus hemipteran1*) negatively affects the reproductive output of its aphid host but crucially induces alary polymorphism, reducing wing development and population crowding, ultimately enhancing individual survival chances [53]. Similarly, a partiti-like virus in the African armyworm (*Spodoptera exempta*) reduces host fecundity while simultaneously enhancing resistance to entomopathogens [54]. These examples collectively illustrate that ISVs exert multifaceted regulatory influences on insect hosts (Figure 2), necessitating comprehensive evaluation across environmental and physiological dimensions.

### 4.2. ISVs Regulation of Arbovirus Transmission

In recent years, ISVs in crucial vector species have attracted considerable research interest. Aside from modulating essential physiological functions of their insect hosts, ISVs exhibit regulatory potential over arboviral replication and transmission, highlighting their potential application in vector-borne disease management [25,55]. To date, research in this field has predominantly focused on mosquito-specific viruses and mosquito-borne pathogens, whereas studies on the regulatory roles of ISVs in plant virus transmission remain limited.

Certain ISVs enhance arbovirus replication and transmission efficiency within their vectors. For example, the peach aphid (*Myzus persicae*), a vector of Potato virus Y (PVY, *Potyvirus yituberosi*), harbors Myzus persicae nicotianae densovirus (MpnDV), which promotes aphid dispersal by regulating host activity-related genes. This behavioral modulation accelerates the systemic spread of PVY in tobacco plants [56]. However, because PVY is transmitted by *M. persicae* in a nonpersistent manner [57], direct viral–viral interactions are unlikely; thus, the promotional effect of MpnDV on PVY transmission is an indirect consequence of altered host behavior. In mosquitoes, the globally distributed Phasi Charoen-like virus (PCLV, *Phasivirus phasiense*) and Humaita Tubiacanga virus (HTV) are prevalent in *Aedes aegypti* populations and are associated with a twofold increase in dengue virus (*Orthoflavivirus denguei*) co-infection in wild mosquitoes. This phenomenon is attributed to the ability of HTV and PCLV to inhibit the downregulation of histone H4, identified in vivo as a critical proviral host factor. Experimental evidence further shows that these ISVs enhance vector competence for the transmission of both dengue and Zika viruses to vertebrate hosts [58]. Some ISVs interact directly with arboviruses. In the small brown planthopper (*L. striatellus*), the capsid protein VP1 of Himetobi P virus (HiPV, *Triatovirus himetobi*) specifically binds to the ribonucleoprotein of Rice stripe virus (RSV, *Tenuivirus oryzaclavatae*), thereby promoting RSV accumulation within the vector [59]. In the leafhopper *Recilia dorsalis*, both the leafhopper-specific recilia dorsalis filamentous virus (RdFV) and Rice gall dwarf virus (RGDV, *Phytoreovirus betaoryzae*) utilize the sperm-specific serpin protein HongrES1 to bind directly to sperm surfaces. This capsid-mediated interaction facilitates the dual invasion of male reproductive organs, increasing paternal vertical transmission rates [51]. Collectively, these examples highlight the diverse strategies by which ISVs influence virus transmission—either indirectly, by modulating host biology, or directly, via protein interactions with co-infecting viruses (Figure 2).

Current evidence supporting ISV-mediated arbovirus suppression primarily focuses on mosquito-borne viruses. For instance, Nhumirim virus (NHUV), a flavivirus closely related to West Nile virus (WNV, *Orthoflavivirus nilense*), significantly inhibits WNV proliferation in Culex mosquitoes and cell cultures [60]. Further investigations have unveiled NHUV′s broad-spectrum antiviral efficacy in C6/36 cells, suppressing various arboviruses, including several flaviviruses (Japanese encephalitis virus, St. Louis encephalitis virus, dengue virus type-2 (DENV-2), and Zika virus (ZIKV, *Orthoflavivirus zikaense*)) and an alphavirus (chikungunya virus) [61,62]. Although research on ISVs′ capacity to suppress plant virus replication and transmission remains limited, emerging evidence has begun to support this phenomenon. For example, an antagonistic relationship has been documented between NcPSRV-1 and RDV in the leafhopper vector [63]. Proposed mechanisms by which ISVs mediate inhibition of plant viruses include: (i) Superinfection Exclusion (SE), a phenomenon in which prior viral infection confers cellular resistance to secondary infection by homologous or related viruses [25,55,64]; (ii) Primed Immune Responses, the initial ISV infection may broadly activate insect immune pathways, establishing cross-protective barriers that restrict subsequent viral invasion [34,55]; (iii) Indirect Physiological Constraints, ISV-induced physiological alterations in insects could indirectly reduce vector competence for pathogen transmission [28]. Although ISVs show great potential for controlling arbovirus transmission, this area of research remains to be further explored.

## 5. Toward Application: Challenges and Innovative Strategies

Historically, researchers have explored the utilization of insect-associated microbes, including viruses and bacteria, for managing agricultural pests and pathogens. Substantial progress has been achieved in targeting animal pathogens transmitted by insect vectors, particularly mosquito-borne diseases [55]. In contrast, strategies against vector-borne plant pathogens remain in nascent developmental stages.

Significant advances have been made in harnessing mosquito-associated symbiotic microbes, including symbiotic bacteria and ISVs, to combat vector-borne pathogens. A notable example is the naturally occurring bacterium Rosenbergiella _YN46, which suppresses dengue and Zika virus infections in Aedes mosquitoes by acidifying the midgut lumen post-blood meal [65]. This symbiont represents a promising candidate for flavivirus biocontrol. Furthermore, genetic engineering of the prevalent symbiont Serratia AS1 enables simultaneous production of anti-Plasmodium and anti-arboviral effector proteins, establishing a dual-pathogen blocking system within mosquito vectors [66]. Parallel strategies guide the application of mosquito-specific viruses. First, identifying wild-type ISVs that impair vector physiology or pathogen transmission (e.g., Nhumirim virus inhibiting WNV replication [61,62]) through laboratory validation. Second, developing recombinant ISVs that either express arbovirus-interfering proteins or trigger RNAi-mediated defenses against target pathogens. Based on the RNA interference (RNAi) pathway innate to insects, targeted gene silencing can be initiated through the delivery of homologous double-stranded RNA (dsRNA) fragments. Since RNA viruses generate dsRNA intermediates during their replication cycle, they represent a theoretically feasible vector for dsRNA delivery [67,68,69]. An early study demonstrated that *Aedes aegypti* mosquitoes injected with engineered Sindbis viruses (*Alphavirus sindbis*) expressing dengue virus-derived sequences exhibited RNAi-mediated immunity against DENV and suppressed viral replication [70]. Similarly, the Flock House virus (FHV, *Alphanodavirus flockense*), an insect virus capable of infecting multiple aphid species, has been genetically engineered. Researchers modified FHV RNA2 to carry foreign gene sequences. Critically, this engineered RNA2 can be encapsidated by FHV virus particles. Upon viral entry into the insect host, the RNAi was triggered against the target gene encoded by the foreign sequence. This establishes a framework for designing precision-engineered ISVs as next-generation biocontrol agents [69]. While the application of ISVs to control plant virus transmission has not yet been reported, this area of research presents considerable potential and promising prospects for future practical implementation.

While modifying ISVs for disease control is theoretically feasible, there are many other considerations and technical difficulties to overcome. Firstly, selecting a suitable candidate virus for modification is crucial. We need a virus that is relatively amenable to modification. Secondly, the inherent biological properties of the modified virus must be carefully evaluated. Key questions include: Is the virus capable of efficient horizontal or vertical transmission within insect populations? Does it adversely affect the insect host itself? These factors are critical as they directly impact the spread and efficacy of the recombinant virus in natural settings or when applied [55,71]. Thirdly, ensuring the safety and genetic stability of the recombinant virus during propagation is paramount. A significant challenge is the tendency for recombinant viruses to lose inserted genetic sequences over time. If this occurs, the virus reverts to its wild-type form, nullifying its intended application. The potential consequences of such reversion must be thoroughly assessed.

## 6. Conclusions

In summary, the widespread application of HTS and continuous refinement of analytical tools have facilitated the discovery of numerous insect-associated viruses. However, current understanding of these viruses remains primarily limited to their genomic sequences, with critical knowledge gaps persisting regarding functional roles, transmission dynamics, virion morphology, and other key characteristics. Given the host specificity of ISVs and their demonstrated ability to modulate insect physiology or influence pathogen transmission efficiency, exploring their potential for biological control applications is of substantial scientific and practical significance. Although these research endeavors are still in early developmental stages, their long-term implications for vector-borne disease management and agricultural pest control are far-reaching.

## Figures and Tables

**Figure 1 viruses-17-01269-f001:**
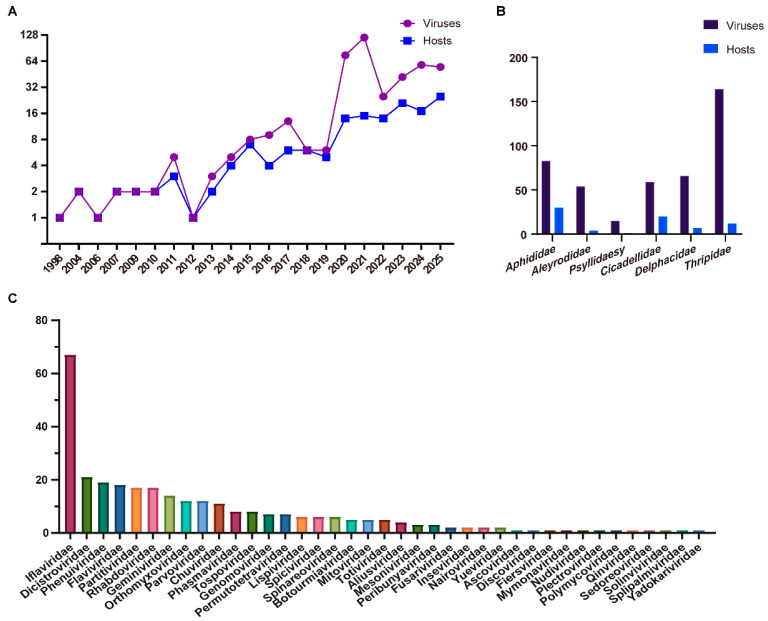
Diversity of viruses in piercing-sucking insects. (**A**) The increasing number of viruses and piercing-sucking insect hosts from 1998 to 2025. (**B**) The distribution of viruses in piercing-sucking insect hosts species across different families. (**C**) The number of viruses belonging to different families.

**Figure 2 viruses-17-01269-f002:**
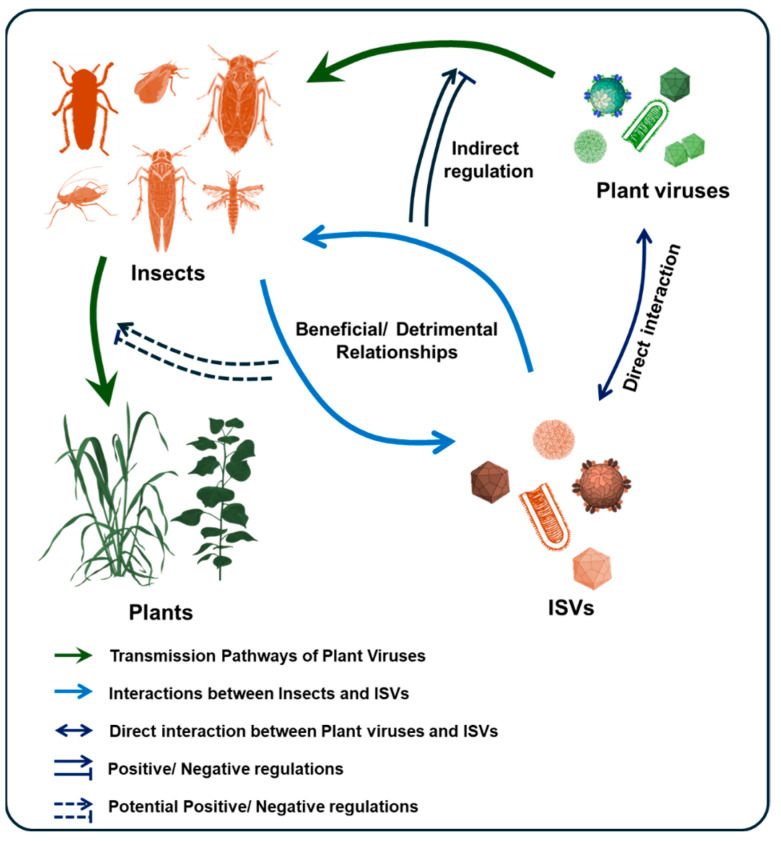
ISVs Regulation of Insect Biology and Virus Transmission. The schematic diagram illustrates the relationship between insects, ISVs, and plant viruses in the process of plant virus transmission. The influence of ISVs on insect hosts may be beneficial or detrimental. ISVs can interact directly with plant viruses or indirectly regulate the replication and transmission of plant viruses in insects by affecting insect biology.

## Data Availability

No new data were created or analyzed in this study. Data sharing is not applicable to this article.

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
