# Peer review of "Insect-Specific Viruses and Their Emerging Role in Plant Disease Mitigation"

_viruses, 2025, doi:10.3390/v17091269_

Round 1

Reviewer 1 Report

Comments and Suggestions for Authors

The paper is interesting and well written, I suggest two main changes:

1- "Next generation sequence" is outdated, since the techonology spread more than 15 years ago. I suggest to modyfy with the High Troughtput Sequence

2- All the viruses name must be corrected. None of them start with the capital letter, and binomial nomenclatura must be added. 

Author Response

Comments 1: "Next generation sequence" is outdated, since the techonology spread more than 15 years ago. I suggest to modyfy with the High Troughtput Sequence

Response 1: Thank you for your suggestions. We have replaced “next generation sequencing (NGS)” with “high-throughput sequencing (HTS)”.

Comments 2: All the viruses name must be corrected. None of them start with the capital letter, and binomial nomenclatura must be added. 

Response 2: We sincerely appreciate this insightful comment. We have conducted a comprehensive check of the entire manuscript to correct the formatting of all virus names. Specifically, we have ensured that the first letter of each virus name is capitalized in accordance with academic writing conventions. Furthermore, referring to the Virus Metadata Resource (VMR_MSL40.v1.20250307) officially released by the International Committee on Taxonomy of Viruses (ICTV), we have supplemented the taxonomic species names for all viruses mentioned in the article.

Reviewer 2 Report

Comments and Suggestions for Authors

This is a well-written, comprehensive review of a new and developing topic. However, the use of the term "plant cancer" on line 27 is problematic and can be misleading as a scientific comparison.  Although they can develop tumor-like growths, plants don't get cancer in the same way animals do. These "plant tumors," such as crown galls, are usually benign and don't spread throughout the plant because their cells have rigid cell walls that prevent them from metastasizing. The growths are typically caused by pathogens, such as the bacterium Agrobacterium tumefaciens, or physical injuries, which lead to uncontrolled cell division. Please remove this idea.

In Line 51, I suggest changing "The efficacy of plant virus transmission varies across populations of various insect species. " to "The efficacy of plant virus transmission differs among various insect populations."

Author Response

Comments 1: This is a well-written, comprehensive review of a new and developing topic. However, the use of the term "plant cancer" on line 27 is problematic and can be misleading as a scientific comparison.  Although they can develop tumor-like growths, plants don't get cancer in the same way animals do. These "plant tumors," such as crown galls, are usually benign and don't spread throughout the plant because their cells have rigid cell walls that prevent them from metastasizing. The growths are typically caused by pathogens, such as the bacterium Agrobacterium tumefaciens, or physical injuries, which lead to uncontrolled cell division. Please remove this idea.

Response 1: Thank you for pointing out this issue. We have revised the sentence to: "Plant viruses pose a significant threat to crop yield and quality, substantially hampering agricultural productivity."

Comments 2: In Line 51, I suggest changing "The efficacy of plant virus transmission varies across populations of various insect species. " to "The efficacy of plant virus transmission differs among various insect populations."

Response 2: Thank you. We have revised this sentence following your suggestions.